# The Potential Use of Metformin, Dipyridamole, N-Acetylcysteine and Statins as Adjunctive Therapy for Systemic Lupus Erythematosus

**DOI:** 10.3390/cells8040323

**Published:** 2019-04-06

**Authors:** Marcus Kai Xuan Tan, Thurston Yan Jia Heng, Anselm Mak

**Affiliations:** 1Yong Loo Lin School of Medicine, National University of Singapore, Singapore 119228, Singapore; e0012286@u.nus.edu (M.K.X.T.); e0012266@u.nus.edu (T.Y.J.H.); 2Department of Medicine, Yong Loo Lin School of Medicine, National University of Singapore, Singapore 119228, Singapore; 3Division of Rheumatology, University Medicine Cluster, National University Health System, Singapore 119228, Singapore

**Keywords:** lupus, SLE, metformin, statin, dipyridamole, acetylcysteine, immunosuppression

## Abstract

Systemic lupus erythematosus (SLE) is a chronic inflammatory autoimmune condition that can potentially affect every single organ during the course of the disease, leading to increased morbidity and mortality, and reduced health-related quality of life. While curative treatment is currently non-existent for SLE, therapeutic agents such as glucocorticoids, mycophenolate, azathioprine, cyclosporine, cyclophosphamide and various biologics are the mainstay of treatment based on their immunomodulatory and immunosuppressive properties. As a result of global immunosuppression, the side-effect profile of the current therapeutic approach is unfavourable, with adverse effects including myelosuppression, infection and malignancies. Hydroxychloroquine, one of the very few Food and Drug Administration (FDA)-approved medications for the treatment of SLE, has been shown to offer a number of therapeutic benefits to SLE patients independent of its immunomodulatory effect. As such, it is worth exploring drugs similar to hydroxychloroquine that confer additional clinical benefits unrelated to immunosuppressive mechanisms. Indeed, apart from hydroxychloroquine, a number of studies have explored the use of a few conventionally non-immunosuppressive drugs that are potentially useful in the management of SLE. In this review, non-immunosuppressive therapeutic agents, namely metformin, dipyridamole, N-acetylcysteine and statins, will be critically discussed with regard to their mechanisms of action and efficacy pertaining to their potential therapeutic role in SLE.

## 1. Introduction

Systemic lupus erythematosus (SLE) is a prototype multi-systemic autoimmune condition characterised by a plethora of autoantibody production, immune complex-mediated inflammation and subsequent organ damage. Although both the short- and long-term survival rates of SLE patients have increased tremendously over the past 50 years, the mortality rate of SLE patients remains 2–4 times higher than that of healthy individuals [1]. In addition, patients with SLE often suffer from substantial morbidity related to the condition and therapies, leading to vocational disability and consequent economic burden to society [2].

Despite the recognition of these negative impacts on patients with SLE, a lack of knowledge regarding the pathophysiology of SLE has led to poor treatment advancement of the condition over the past four to five decades. Currently, the mainstay of therapeutic management of SLE is largely based on immunosuppression, involving the use of glucocorticoids and other global immunosuppressive (e.g., mycophenolate, cyclophosphamide, cyclosporin) or biologic agents (e.g., rituximab, belimumab) [1]. However, the immunosuppressive nature of these drugs predisposes patients to a number of potential major side effects, including the risk of myelosuppression, infections and malignancies [3]. Moreover, novel biologic agents, which are designed to be more selective in their immunosuppression mechanism, are not spared from major side effects, including hypogammaglobulinemia and progressive multifocal leukoencephalopathy related to rituximab [4,5]. Furthermore, clinical trials on the use of these biologic agents in SLE have yielded mixed results [6], with non-inferiority trials such as the EXPLORER and LUNAR trials on rituximab failing to reach their respective primary endpoints [7,8]. On the other hand, the BLISS-52 and BLISS-76 trials on belimumab (the first and only FDA-approved biologic agent for SLE treatment) have shown potentially promising results by reaching a more sophisticated primary endpoint [9,10].

Antimalarial drugs including chloroquine and hydroxychloroquine (HCQ), one of the very few categories of medications that was approved by the FDA for the treatment of SLE more than 60 years ago, have been shown to confer a number of therapeutic benefits to SLE patients [11]. Aside from its immunomodulatory effects, including interruption of antigen presentation, Toll-like receptor (TLR) signalling and formation of neutrophil extracellular traps (NETs) that have a substantial impact on the pathophysiology of SLE [12,13,14], antimalarials have been demonstrated to possess lipid- and sugar-lowering properties, as well as an anti-thrombotic effect [15,16]. Additionally, it has recently been demonstrated that HCQ significantly improves the health-related quality of life in patients with lupus nephritis [17]. As such, it is worth exploring therapeutic agents similar to HCQ that are both safe and capable of offering additional therapeutic benefits independent of their immunosuppressive mechanisms in the treatment of SLE. Indeed, apart from HCQ, a few conventionally non-immunosuppressive drugs that are potentially useful in the management of SLE have been intensively studied.

In this paper, we review the studies that address the potential therapeutic benefits of four non-immunosuppressive drugs in the management of SLE; namely metformin, dipyridamole, N-acetylcysteine and statins. Along with HCQ, these four drugs have been studied extensively in the field of adjunctive SLE therapy, with many studies showing their potential therapeutic benefits. In contrast to steroids and other immunosuppressive drugs used to manage SLE, which have been shown to compromise the host’s immune system, these four drugs have been found to cause alterations to the immune system, without any evidence suggesting that they compromise the potency of the immune system of the host. Hence, they may potentially have a better side effect profile than current immunosuppressive drugs. Studies investigating the use of these four drugs ranged from laboratory studies on different lupus-prone mouse models to randomised clinical trials on human subjects. Notably, these medications play an important role in modulating the molecular mechanisms involved in the pathogenesis of SLE, leading to potentially significant therapeutic benefits pertaining to reduction of SLE-related disease activity, damage and co-morbidities.

## 2. Methods

### 2.1. Literature Search

This systematic review was conducted in accordance with the guidelines of Preferred Reporting Items for Systematic Reviews and Meta-Analyses (PRISMA) [18]. A literature search for publications published between 1966 and March 2019 was conducted in the electronic database PubMed. To identify potentially relevant publications, a combination of keywords including the names of each drug (“metformin”, “dipyridamole”, “acetylcysteine” and “statin”) and “sle”, “lupus” or “systemic lupus erythematosus” were used. The search was conducted using the following search term, which was repeated for each of the four drugs: “(name of drug) AND (sle OR lupus OR systematic lupus erythematosus)”.

### 2.2. Study Eligibility Criteria

Articles published between 1966 and March 2019 that investigated the mechanisms of action and efficacy of the relevant four drugs with regard to their potential therapeutic use in SLE were included. Exclusion criteria of articles included non-English publications, articles irrelevant to the treatment of SLE or with insufficient information regarding the drugs of interest, case reports or series, review papers, editorials or comments and articles addressing specific subject groups (e.g., pregnant women).

## 3. Results

The initial search identified a total of 278 hits. Duplicate records (n = 4) were removed. During the screening stage, 144 papers (non-English publications (n = 33), case reports or case series (n = 39), review articles (n = 55) and editorials or letters (n = 17)) were excluded. As for the subsequent assessment of 130 full texts, 100 of them (irrelevant to SLE treatment (n = 66), insufficient information regarding mechanism of action of drug (n = 25) and articles only relevant to specific groups (n = 9)) were excluded. Finally, 30 papers were included for discussion in this review (see Figure 1 for the summary of literature search).

## 4. Discussion

### 4.1. Metformin

Metformin is a biguanide oral hypoglycaemic agent that is widely used as a first-line pharmacological therapy of type II diabetes mellitus [19]. It functions primarily via enhancing insulin sensitivity, reducing hepatic glucose production and increasing glucose uptake by peripheral tissues [20,21]. Metformin is a generally safe drug, which does not typically result in hypoglycaemia. Common adverse effects include gastrointestinal intolerance (with symptoms including nausea, vomiting and diarrhoea) and a metallic taste in the mouth [22]. Although rare, increased risk of lactic acidosis is well known with the use of metformin, particularly in patients with renal and heart failure, sepsis and hypoxia [23]. In addition to type II diabetes mellitus, metformin is used in the management of various other conditions including obesity and polycystic ovarian syndrome [24]. Recently, a number of mechanistic studies have identified potential benefits of metformin in the treatment of SLE [12,25,26,27].

#### 4.1.1. Inhibition via AMPK/mTOR/STAT3 Pathway

Metformin was able to regulate various cellular differential processes and signalling, such as inhibition of the differentiation of B cells into antibody-producing plasma cells and differentiation of Th17 via the AMP kinase/mechanistic target of rapamycin/signal transducer and activator of transcription 3 (AMPK/mTOR/STAT3) pathway [25]. The AMPK and mTOR pathways are involved in the regulation of cell growth [28] and epigenetic modifications including methylation and histone phosphorylation, leading to enhancement of T cell activation [29]. Metformin activated AMPK via inhibition of mitochondrial complex I, leading to an increase in the AMP/ATP or ADP/ATP ratio [30]. This prevented the phosphorylation and subsequent activation of mTOR, a downstream target of AMPK and caused the inhibition of B cell proliferation and formation of germinal centres [31]. Similarly, with mTOR inhibition, signalling did not proceed via mTOR complex 1 (mTORC1). This lifted the inhibition of Foxp3 which resulted in activation of T regulatory (Treg) cells. STAT-3 activation via phosphorylation was also prevented [32], indirectly suppressing the differentiation of Th17 and B lymphocytes, and hence the production of autoantibodies [33,34]. Of note, the p53 expression was found to be markedly increased in CD4^+^ T cells of metformin-treated *Roquin^san/san^* mice (a murine model of SLE), which has been shown to directly suppress Th17 differentiation by interacting with STAT-3 [35]. The use of metformin has been shown to decrease the serum levels of anti-dsDNA, total IgG and serum IgG1 in *Roquin^san/san^* mice [25]. A marked reduction in ICOS-expressing T follicular helper (Tfh) cells (which are responsible for maintaining and regulating the differentiation of germinal centre B cells) was also demonstrated in the spleen of metformin-treated *Roquin^san/san^* mice, which is coupled with a decreased population of GL-7-expressing germinal centre B cells [25].

#### 4.1.2. Inhibition of Oxidative Phosphorylation

Metformin was found to normalise CD4^+^ T-cell glucose metabolism via inhibition of mitochondrial complex I and oxidative phosphorylation [26]. Glucose metabolism was crucial for the activation, proliferation and differentiation of CD4^+^ T cells [36], contributing to the patho-immunological response in SLE via the production of overactive T effector cells (including Th1 and Th17 cells) and proinflammatory cytokines (including interferon (IFN)-γ and interleukin (IL)-17). Enhanced lactate production and mitochondrial oxygen consumption were also observed in CD4^+^ T cells in lupus-prone B6.*lpr* mice and SLE patients. Besides decreasing oxygen consumption in T cells, metformin was found to normalise the excessive IFN-γ production by CD4^+^ T cells [27]. It has been shown that monotherapy with metformin prevented the development of SLE through reduction in CD4^+^ T cell activation, expansion of germinal centre Tfh and B cells, serum antinuclear antibody (ANA) level and severity of lupus nephritis. While metformin alone may prevent the activation of autoimmune CD4^+^ T cells, the combination with 2-deoxy-D-glucose (2DG) was required to normalise chronically activated CD4^+^ T cells [27]. The dual-therapy was also effective in suppressing autoimmune manifestations and reducing the number and percentage of splenic Tfh cells in lupus-prone mice [26]. However, the therapeutic benefit of the dual-therapy rapidly disappeared following cessation of treatment, showing that continuous exposure to both therapeutic agents is required to downregulate the activation and effector differentiation of the pathogenic CD4^+^ T cells.

#### 4.1.3. Downregulation of NET mtDNA-PDC-IFNα Pathway

Metformin was found to downregulate the NET mitochondrial DNA-plasmacytoid dendritic cells-type I interferon-α (mtDNA-PDC-IFNα) pathway [12]. Metformin use led to a significant reduction in NET DNA release from neutrophils via a decrease in reactive oxygen species (ROS) production, as a result of the ability of metformin to selectively inhibit mitochondrial complex I, which reduces the activity of NADPH oxidase [37]. Similarly, metformin has been shown to decrease the number of mtDNA copies in NET and inhibit the CpG or mtDNA/anti-mtDNA autoantibody-stimulated IFN-α generation in a dose-dependent manner by plasmacytoid dendritic cells. This is crucial, given that mtDNA has been shown to be more efficacious in activating IFN-α as compared to dsDNA in NETosis. The anti-mtDNA auto-antibodies were also shown to have a stronger correlation, compared to anti-dsDNA, with the quantity of proteinuria and lupus nephritis activity index (according to the International Society of Nephrology and the Renal Pathology Society 2003 classification criteria) in SLE patients [12]. In addition, the use of metformin in SLE patients has been found to reduce the risk of disease flares (assessed by the SLE Disease Activity Index (SLEDAI) and modified Safety of Estrogens in Lupus Erythematosus National Assessment version of the SLEDAI Flare Index (SFI)) by 51%, as well as subsequent reduction of glucocorticoid dose [12]. The use of metformin in SLE patients beyond eight weeks also resulted in a statistically significant decrease in body mass index (BMI), with a possible improvement in cardiovascular risk and wellbeing of the patients [12].

### 4.2. Dipyridamole

Dipyridamole is a phosphodiesterase inhibitor, which primarily functions to increase intracellular cAMP levels and block adenosine reuptake in cells, resulting in both antiplatelet and vasodilatory effects [38,39,40]. It is commonly used as an antithrombotic agent and a pharmacological agent in stress radionuclide myocardial perfusion imaging. Interestingly, it has been shown that dipyridamole possesses anti-inflammatory effects. It is hence able to act synergistically with glucocorticoids, making it a potential candidate drug to be added to the SLE treatment regime [41].

#### Inhibition of Calcineurin/NF-AT Pathway

Lupus T cells were found to display substantial calcium flux and a hyperactive calcineurin/nuclear factor of activated T cells (NF-AT) pathway, resulting in an increased transcription of genes (e.g., CD154, a costimulatory molecule) for T cell-directed autoantibody production by B cells in SLE patients [41]. Through inhibition of the calcium-dependent signalling events in the calcineurin/NF-AT pathway, dipyridamole was found to reduce the activation and proliferation of double-negative T cells (DNTC) and associated production of the proinflammatory cytokines IFN-γ, IL-6 and IL-17 in lupus-prone MRL/*lpr* mice. Dipyridamole was also found to decrease T cell-dependent immunoglobulin production by B cells by more than 50% [41]. However, unlike the calcineurin inhibitor tacrolimus, dipyridamole does not lead to B cell activation *per se*. In lupus-prone MRL/*lpr* mice, the onset of lupus-related pathology, such as nephritis and skin disease, was delayed compared to control MRL/MpJ mice [41]. Nevertheless, further studies are required to define the role of and timing of dipyridamole administration to better understand the pharmacodynamic or pharmacokinetic characteristics and their effects on immune homeostasis.

### 4.3. N-Acetylcysteine

N-acetylcysteine (NAC) is an antioxidant that functions to scavenge free radicals and enhance glutathione biosynthesis and detoxification [42]. It is potentially therapeutic for diseases in which the impact of free oxygen radicals on the disease pathophysiology is substantial [43]. This includes treatment for acetaminophen toxicity [44], chronic bronchitis [42] and obstetric and gynaecological conditions such as polycystic ovary syndrome [45] and premature birth [43].

In recent years, NAC has been shown to be potentially beneficial to the treatment of SLE with a favourable side-effect profile as a result of its effects in the immune system [46,47,48,49,50,51,52,53]:

#### 4.3.1. Inhibition of mTOR Pathway

NAC was found to reduce disease activity in SLE patients by disrupting the mTOR pathway in DNTC [46], in a fashion reminiscent of the pharmacological action of rapamycin [54]. Being a scavenger of ROS and a precursor of glutathione, NAC decreased levels of ROS while increasing the levels of glutathione in peripheral lymphocytes. Increasing levels of glutathione resulted in sparing of NADPH (since less NADPH is required for the reduction of GSSG to GSH), which may explain the increase in de novo nitric oxide (NO) synthesis [55], leading to a decrease in activation of the mTOR pathway due to changes in mitochondrial hyperpolarization. An increase in FoxP3 expression in CD4^+^CD25^+^ regulatory T cells was also noted with NAC use. At the same time, apoptosis of DNTC, a subset of leucocytes that promotes autoantibody production [56], was significantly enhanced by NAC. Additionally, the expansion of DNTC was effectively reversed, and serum anti-DNA levels were decreased.

Kynurenine, a metabolite of the pentose phosphate pathway, serves as a metabolic checkpoint in the pathogenesis of SLE in DNTC (which are a source of IL-4, IL-17 and necrotic debris) [57]. Accumulation of kynurenine was found to play an important role in the activation of mTOR in SLE [47]. Treatment with NAC led to the sparing of NADPH, resulting in increased catabolism by NADPH-dependent kynurenine hydroxylase. The resultant significant reduction in kynurenine levels inhibited the mTOR pathway in DNTC of SLE patients [47]. A separate study on SLE patients found that NAC therapy also led to the blockade of electron transport chain (ETC) complex I activity and a reduction in H_2_O_2_ production (which could be due to the antioxidant effects of this blockade) [48]. The inhibition of ETC complex I, which is crucial for oxygen consumption in peripheral blood lymphocytes in SLE patients, may hence explain the improvement in disease activity with treatment with NAC.

#### 4.3.2. Reversal of Effects of ROS on F-Actin Polymerisation

SLE patients were found to have reduced levels of superoxide dismutase (SOD), an antioxidant enzyme involved in scavenging intracellular ROS, thereby increasing ROS levels [49]. Consequently, the increase in ROS impeded the polymerisation of F-actin and impaired the migration and homing capacity of bone marrow mesenchymal stem cells (MSCs) [58]. NAC, being an antioxidant and scavenger of ROS, was found to reverse the above process in SLE patients. NAC was also shown to significantly reduce the serum levels of anti-dsDNA antibodies and improve lupus nephritis in lupus-prone MRL/*lpr* mice [58]. A separate study on female NZB x NZW F1 (B/W) mice (a mouse model of SLE) similarly revealed a decrease in ROS and anti-dsDNA antibody levels after treatment with NAC, leading to a survival benefit as compared to their control counterpart [50].

#### 4.3.3. Opposing the Effects of Trichloroethene

Treatment with NAC was found to oppose the mechanisms of trichloroethene (TCE), an environmental contaminant commonly involved in the pathogenesis of a number of autoimmune diseases such as SLE, scleroderma and autoimmune hepatitis via oxidative and nitrosative stresses [51]. TCE exposure resulted in decreased glutathione levels, leading to an increase in ROS levels which contribute to oxidative stress. Increased levels of nuclear factor kappa-light-chain-enhancer of activated B cells (NF-κB) p65 and inducible nitric oxide synthase (iNOS), which are both involved in the production of reactive nitrogen species-modified (RNS-modified) proteins such as nitrotyrosine, were noted. RNS-modified proteins can act as immunogens or neoantigens, potentially activating autoreactive B and T cells (especially the Th1 subset), which are substantially involved in the initiation and perpetuation of autoimmunity [59]. TCE-exposed mice also demonstrated elevated serum levels of ANA and anti-histone antibody (AHA) levels. Treatment with NAC was found to increase intracellular glutathione levels and decrease the levels of the proinflammatory nitrotyrosine, NF-κB p65, iNOS and autoantibodies [51].

#### 4.3.4. Improvement of Endothelial Dysfunction

Patients with SLE have increased cardiovascular risk and mortality [60]. Endothelial dysfunction, which is one of the initial steps involved in the pathogenesis of atherosclerosis, has been demonstrated in SLE patients [61]. Thus far, there has been no therapeutic agent that addresses endothelial dysfunction in patients with SLE.

Apart from the aforementioned beneficial effects, NAC has also been shown to improve endothelial dysfunction in SLE patients [52]. Treatment with NAC has been shown to improve endothelial dysfunction, as shown by a reduction in reflective index, a biophysical measurement of endothelial dysfunction and arterial stiffness. Administration of NAC can also lead to a reduction in C-reactive protein (CRP) and serum malondialdehyde (MDA) levels, which are markers of inflammation and oxidative stress, in patients with SLE [52]. In addition, scavenging of ROS by NAC was also noted to abort osteoprotegerin-induced apoptosis of endothelial progenitor cells, which was strongly correlated with endothelial cell dysfunction in patients with SLE [53].

### 4.4. Statins

Statins are a class of lipid-lowering drugs, typically used in the treatment of hypercholesterolemia and its cardiovascular-associated diseases [62]. They chiefly function by inhibiting 3-hydroxy-3-methylglutaryl coenzyme A (HMG-CoA) reductase.

Statins lower low-density lipoproteins (LDL) and serum CRP levels, offering primary and secondary prevention against cardiovascular events [63,64]. Beneficial vascular effects include a decrease in arterial stiffness, vascular endothelial growth factor (VEGF) and soluble vascular cell adhesion molecule-1 (sVCAM-1) [65], oxidation status and vascular inflammation [66], as well as the risk of thrombosis [67]. Statins have also been demonstrated to reduce SLE-related disease activity (as measured by SLEDAI scores) [66] and mortality among hyperlipidaemic SLE patients [68]. Additionally, pleiotropic immunomodulatory effects have recently been demonstrated in the use of statins, rendering them potentially useful in the pharmacological management of SLE.

#### 4.4.1. Modulation of Cytokine Levels

Statins have been found to reduce the production of proinflammatory cytokines, particularly IL-17 and IL-21, in SLE patients by inhibiting the Rho-associated, coiled-coil-containing protein kinase (ROCK) pathway [69]. Dysregulated ROCK activity was found in the peripheral blood mononuclear cells of patients with SLE. By inhibiting HMG-CoA reductase, simvastatin has been shown to disrupt the activation of RhoA (a major ROCK activator), and hence inhibited the activation of ROCK. In the absence of phosphorylation by ROCK, production of interferon regulatory factor (IRF)-4 was reduced, resulting in a decrease in the in vitro production of IL-17 and IL-21 by Th17 cells.

Levels of inflammatory cytokines have been found to be altered in gld.apoE^−/−^ mice (mice with lupus-like features and atherosclerosis, which have absent functional Fas ligand and apolipoprotein E) treated with statins [70]. This was due to an increase in STAT6 activation, promoting a Th2 immune response and production of anti-inflammatory cytokines (IL-4 and IL-10) [71,72]. A concomitant decrease in STAT4 activation led to a reduction in Th1 immune response and production of the proinflammatory tumour necrosis factor (TNF)-α and IFN-β [73,74] in the gld.apoE^−/−^ mice. Additional SLE-alleviating effects that have been demonstrated in the gld.apoE^−/−^ mice include reductions in apoptotic debris in lymph nodes, lymphoproliferation, ANA production and proteinuria. Of particular note, although the area of atherosclerotic lesions was reduced, serum cholesterol levels remained unaffected.

In a separate study on SLE patients, a decrease in serum chemokines and cytokines (including IL-6 and IL-8), with an increase in mitochondrial biogenesis and ROS levels, was shown [66]. The same study noted a concurrent alteration in the expression of genes involved in multiple pathways such as inflammation and oxidative stress in monocytes [66]. Such observations showcase that the anti-inflammatory benefits of statins are beyond their lipid-lowering ability.

Interestingly, statins were found to be able to reverse the lipid raft-associated signalling abnormalities in autoreactive T cells from SLE patients [75]. Within lipid rafts, the dynamic localisation of lymphocyte-specific protein tyrosine kinase (Lck) and the linker for activation of T cells (LAT) are potentially crucial for regulating the activation of T cells [76]. However, CD45 is abnormally located within lipid rafts in lupus T cells [77], where it abnormally associates with and increases Lck expression levels. Statins therefore functioned via dissociating CD45 from Lck, hence reducing Lck activation and re-establishing normal levels of Lck expression. In addition, rapid formation of immune synapses has been demonstrated to be reduced as statins disrupted the upregulation of membrane lipids and cholesterol. At the same time, statins also reduced the production of IL-6 and IL-10 in vitro in T cells from SLE patients [76]. Another study on SLE patients found a decrease in plasma chemokine ligand 9 (CXCL9) levels upon treatment with atorvastatin [78]. Expression of CXCL9 on T lymphocytes was induced by IFN-γ, which was found on atherosclerotic lesions. Increased CXCL9 expression resulted in additional recruitment and retention of T lymphocytes in the atherosclerotic lesions [79,80]. Treatment with statins, via its effects on CXCL9, was correlated with a reduction of SLE disease activity as reflected by the overall reduction in the SLEDAI score [78].

Furthermore, statins were noted to inhibit the production of IFN-α and TNF-α by PDCs in SLE patients, via its effect on several pathways [81], including reduction in synthesis of mevalonate (the downstream metabolite of HMG-CoA) and inhibition of CpG-induced phospho-p38 mitogen-activated protein kinase (MAPK) expression. Statins also inhibited the phosphoinositide (PI) 3-kinase pathway, as shown by the reduced expression of phospho-Akt. Similarly, inhibition of the IRF-7 nuclear translocation in PDCs by statins was noted. This is further supported by the slight blockage of TLR-induced NF-κB phosphorylation [81]. In addition, statins were found to inhibit both geranylgeranyl transferase and Rho kinase, thus inhibiting interferon production in a cholesterol-independent manner. The effect of statins on these pathways all contributed to the reduction in production of type 1 IFN by PDCs.

Notably, statins were also found to decrease the soluble TNF-α receptor type 1 (sTNFR1) in SLE patients as compared to controls [82]. Proinflammatory cytokines, such as TNF-α, augmented lipolysis with a resultant rise in free fatty acids [83], contributing to endothelial dysfunction via reducing endothelial NO production and increasing ROS [84]. Since sTNFR1 is involved in anti-apoptotic/inflammatory signalling [85], a reduction in the receptor level potentially ameliorates endothelial dysfunction in SLE patients [82]. However, it is important to note in the same study that the expression of TNF-α and soluble TNF-α receptor type 2 (sTNFR2) did not alter after eight weeks of treatment with atorvastatin. Nevertheless, an independent positive correlation between levels of sTNFR1 and SLEDAI as well as SLE-related organ damage as assessed by the SLICC/SDI index was demonstrated [82], suggesting that statin use potentially benefits SLE patients by reducing SLE-related disease activity and organ damage via its effects on sTNFR1 reduction.

#### 4.4.2. Regulation of TLR Signalling

Statins are capable of increasing the level of soluble Toll-like receptor 2 (sTLR2) [86], an endogenous regulator of TLR signalling, possibly by inhibiting the Rho pathway [87]. TLR2 enhanced autoantibody generation through the activation of various antigen presenting cells including macrophages and dendritic cells via association between nucleosomes and high mobility group box 1 (HMGB1) proteins [88,89]. By acting as a decoy receptor, sTLR2 prevented ligands from binding to TLR2. In addition, by possibly interacting with the coreceptor (CD14) [90], sTLR2 disrupted the close proximity between coreceptor and TLR2, which was required for highly efficient signalling of TLR2. These resulted in a reduction in efficiency of TLR2 signalling. Levels of sTLR2 were also negatively correlated with SLEDAI scores, LDL levels and left ventricular diastolic dysfunction in SLE patients.

#### 4.4.3. Modulation of MHC Class II Expression

Apart from reducing serum cholesterol level, atorvastatin use has been found to lead to reductions of anti-dsDNA IgG level and proteinuria in lupus-prone NZB/W F1 mice [91]. This occurred via down-modulation of major histocompatibility complex class II (MHC-II) expression on monocytes and B cells, which may also possibly explain the reduction in T cell proliferation.

#### 4.4.4. Studies Portraying Contrasting Evidence

Nevertheless, not all studies support the pleiotropic effects of statins in the treatment of SLE. A study performed on lupus-prone LDLr^−/−^ mice revealed that statins reduced cholesterol levels but not atherosclerotic lesion size in the LDLr^−/−^ mice reconstituted with bone marrow from SLE-susceptible B6.Sle1.2.3 mice [92]. In another study, despite the increased serum levels of IL-4 in NZB/NZW mice that reflected Th2 activation in vivo, treatment with statins was not associated with a significant improvement in clinical and laboratory parameters of SLE disease activity including proteinuria, serum anti-dsDNA antibody levels and survival [93]. Human studies addressing the effect of statins on SLE disease activity have equally been non-conclusive [63,94]. The lack of effect of statins on atherosclerotic plaque size in SLE patients was demonstrated [95]. A separate study on SLE patients found a reduction in SLE-related disease activity, as assessed by the Systemic Lupus Activity Measure-Revised (SLAM-R) scores, upon treatment with statins [96]. However, no alterations in pro-inflammatory biomarkers, including IL-1β, IL-6 and IL-8, as well as the prothrombotic antiphospholipid antibodies were noted [96].

## 5. Conclusions

In summary, the potential therapeutic benefits of conventionally non-immunosuppressive drugs, namely metformin, dipyridamole, N-acetylcysteine and statins have been well demonstrated in various experimental, mechanistic, observational and clinical studies reviewed in this article with regard to the management of SLE (see Figure 2 and Table 1 for the summary of these mechanisms).

However, many of these studies included in this review were based on animal studies. Given the limited number of human studies available, the benefits of these drugs may not be directly translatable when used on humans. Multiple studies on statins and their therapeutic benefits in SLE have also yielded mixed conclusions. Nevertheless, while not studied vigorously, the favourable impact of statins and N-acetylcysteine on patients with SLE in the molecular perspective has been preliminarily demonstrated [46,47,94,97].

Yet, at the time of writing of this article, no placebo-controlled trial that aims to address the use of metformin and dipyridamole in SLE has been published. In the non-lupus population, metformin and dipyridamole are often used in combination, particularly in patients with a history of type II diabetes and cerebrovascular diseases, for secondary and tertiary prevention against vascular events [98]. As patients with SLE have increased cardiovascular risk and incidence of vascular accidents compared to demographically-matched healthy subjects [99], the use of metformin and an antiplatelet agent (e.g., dipyridamole) in combination should be considered in diabetic SLE patients who are indicated for secondary cardiovascular protection. As to whether the use of such therapeutic combination can benefit patients in terms of alleviation of SLE-related disease activity and damage besides cardiovascular indications, further observational studies and clinical trials will be necessary to address this interesting yet practical research question.

Undoubtedly, more has to be performed to evaluate the pharmacodynamic and pharmacokinetic properties, adverse effects and ultimate clinical benefits of these commonly-used therapeutic agents before recruiting them into the auxiliary treatment armamentarium of SLE. Given the vast complexity of the disease and its treatment, researchers should collaborate extensively with bedside clinicians to generate more applicable end points for further clinical trials. There remains large potential to further improve the stagnant morbidity and mortality rates in SLE patients in the era of unparalleled advancement of the knowledge of the newer generation of immunosuppressive agents such as biologics and small molecules.

## Figures and Tables

**Figure 1 cells-08-00323-f001:**
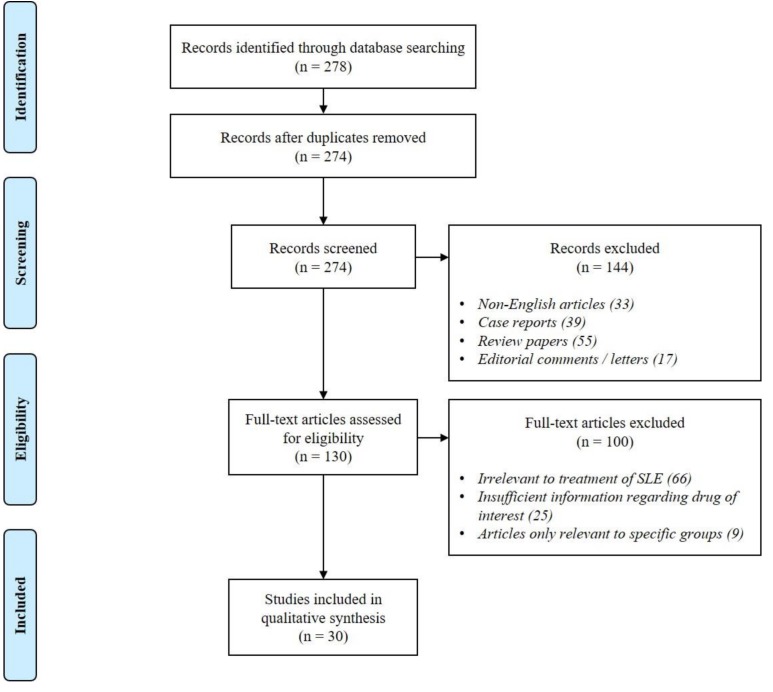
Summary of literature search.

**Figure 2 cells-08-00323-f002:**
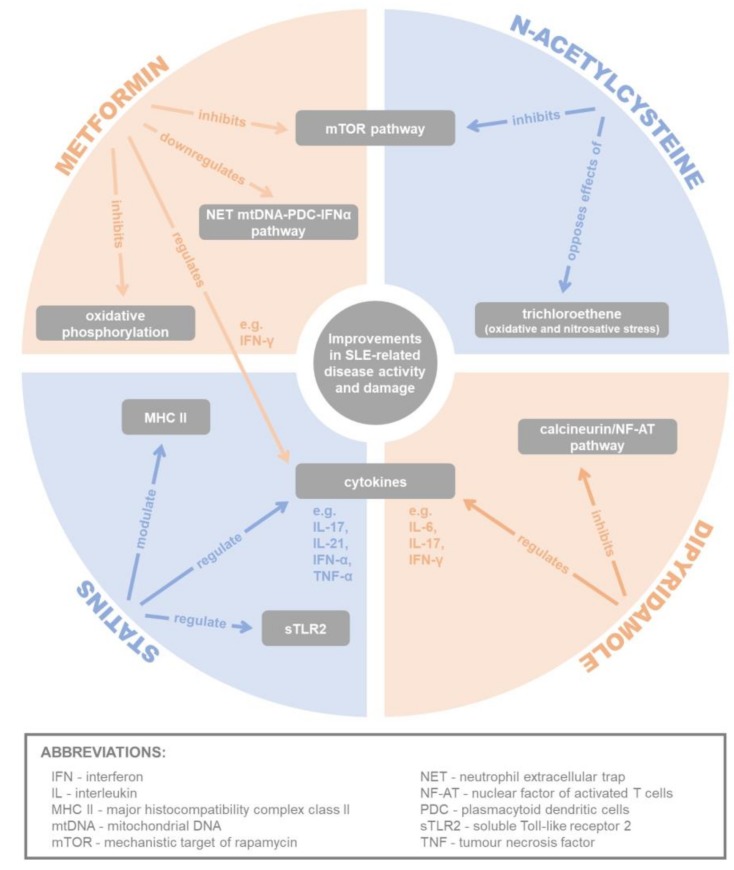
Summary of potential mechanisms of metformin, dipyridamole, N-acetylcysteine and statins in alleviating systemic lupus erythematosus (SLE)-related disease activity and damage.

**Table 1 cells-08-00323-t001:** Summary of studies addressing the potential mechanisms of metformin, dipyridamole, N-acetylcysteine and statins in alleviating SLE-related disease activity and damage.

Author(s) [Ref]	Year	Type of Study	Summary of Mechanism/Results
**Metformin**			
Yin, Y.; et al. [27]	2015	Animal study	Results in reduction in germinal centre B cells, severity of renal pathology and terminal serum ANAs Combination with 2DG is required to normalise chronically activated CD4^+^ T cells
Yin, Y.; et al. [26]	2016	Animal study	Normalises glucose metabolism of CD4^+^ T cells by inhibiting mitochondrial complex I and oxygen phosphorylation Reduces activation, proliferation and differentiation of CD4^+^ T cells
Lee, S.Y.; et al. [25]	2017	Animal study	Inhibits differentiation of B cells Inhibits production of Th17 via AMPK/mTOR/STAT3 pathway Increases p53 expression, which directly suppresses Th17 differentiation Decreases serum levels of anti-dsDNA, total IgG and serum IgG1
Wang, H.; et al. [12]	2015	Human study (NCT02741960; active, not recruiting)	Downregulates NET mtDNA-PDC-IFNα pathway Decreases NET DNA release via decrease in ROS production Decreases number of mtDNA copies in NET and inhibits CpG or mtDNA/anti-mtDNA autoantibody-stimulated plasma dendritic cell IFN-α
**Dipyridamole**			
Kyttaris, V.C.; et al. [41]	2011	Human study (NCT01781611; recruiting)	Inhibits calcium-dependent signalling events in the calcineurin/NF-AT pathway Prevents activation and proliferation of SLE T cells, along with production of proinflammatory cytokines Decreases T cell-directed B cell immunoglobulin production Delays lupus-related pathology
**N-Acetylcysteine**			
Suwannaroj, S.; et al. [50]	2001	Animal study	Decreases ROS and anti-dsDNA antibody levels Prolongs survival
Kudaravalli, J.; et al. [52]	2011	Animal study	Decreases CRP and MDA levels Improves endothelial dysfunction
Kim, J.Y.; et al. [53]	2013	Animal study	Aborts osteoprotegerin-induced apoptosis of endothelial progenitor cells
Wang, G.; et al. [51]	2014	Animal study	Reduces TCE-induced nitrosative stress Increases levels of glutathione Decreases levels of iNOS, NF-κB p65, nitrotyrosine and autoantibodies (ANA, AHA)
Shi, D.; et al. [58]	2014	Animal and human study	Decreases intracellular ROS Results in improvements in polymerisation of F-actin and migration and homing capacity of bone marrow MSCs Significantly reduces serum autoantibody levels and improves lupus nephritis
Lai, Z.W.; et al. [46]	2012	Human study (NCT00775476; suspended—funds exhausted)	Decreases levels of ROS with increased levels of glutathione in peripheral lymphocytes, resulting in sparing of NADPH Decreases activation of mTOR pathway in T cells due to changes in mitochondrial hyperpolarization Decreases DNTC and hence anti-dsDNA levels Increases FoxP3 expression in CD4^+^CD25^+^ T cells
Doherty, E.; et al. [48]	2014	Human study	Blocks ETC complex I activity, with reduction in H_2_O_2_ production
Perl, A.; et al. [47]	2015	Human study (NCT00775476; suspended—funds exhausted)	Spares NADPH, resulting in increased catabolism by kynurenine hydroxylase Decreases kynurenine levels and hence inhibits mTOR pathway
**Statins**			
Lawman, S.; et al. [91]	2004	Animal study	Down-modulates MHC-II expression on monocytes and B cells Reduces serum IgG anti-dsDNA antibodies and proteinuria
Aprahamian, T.; et al. [70]	2006	Animal study	Increases in STAT6 and decrease in STAT4 activation Reduces inflammatory cytokines (including IL-4 and IL-10) Decreases apoptotic debris in lymph nodes, lymphoproliferation, ANA production and proteinuria
Jury, E.C.; et al. [75]	2006	Human study	Reduces Lck activation, restoring Lck expression to normal levels Reduces rapid formation of immune synapses Reverses lipid raft-associated signalling abnormalities Reduces IL-6 and IL-10
Ferreira, G.A.; et al. [78]	2010	Human study	Decreases plasma CXCL9
Amuro, H.; et al. [81]	2010	Human study	Reduces synthesis of mevalonate and inhibition of CpG induced phosphor-p38 MAPK expression Inhibits PI3-kinase pathway and IRF-7 nuclear translocation in PDCs Inhibits geranylgeranyl transferase and Rho kinase Reduces type 1 IFN-α and TNF-α by PDCs
Ruiz-Limon, P.; et al. [66]	2015	Human study	Decreases serum cytokines and chemokines (e.g., IL-6, IL-8) Increases mitochondrial biogenesis and reduces ROS levels
Ferreira, G.A.; et al. [82]	2016	Human study	Decreases sTNFR1, which is involved in anti-apoptotic/inflammatory signalling Positively correlates between sTNFR1 and SLEDAI, as well as SLICC
Houssen, M.E.; et al. [86]	2016	Human study	Increases levels of sTLR2 possibly by inhibiting Rho pathway Reduces efficiency of TLR2 signalling
Rozo, C.; et al. [69]	2017	Human study	Reduces production of interleukins (IL-17, IL-21) by disrupting activation of RhoA and inhibiting ROCK pathway

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
