# Peer review of "The Potential Use of Metformin, Dipyridamole, N-Acetylcysteine and Statins as Adjunctive Therapy for Systemic Lupus Erythematosus"

_cells, 2019, doi:10.3390/cells8040323_

Round 1

Reviewer 1 Report

Several extensive reviews and monographs have appeared in the recent years on new therapies for SLE, and this represents a lot to read for beginners in the field. With regard to these aspects, this comprehensive review article, which is specially focused on non-immunosuppressive therapies, is welcome. Quite regrettably, however, the presentation of text is sometimes superficial, with typical sentences that can be read in many other reviews and articles (especially in the Introduction) without any critical thinking. The tone that is set remains essentially descriptive and the authors express no opinion or more personal ideas. They do not mention any limitations or criticisms in the analyses (shortly evoked in lines 360-365 of the Conclusion). These aspects are important in a review article because they can orientate future research.   

Specific comments

1. The title is misleading as four so-called “non-immunosuppressive” molecules only are described in this article. This partial choice of selected tools is fully acceptable but this should be mentioned in the title.

2. The word “including” (lines 25 and 357) should be deleted since no other molecules than these 4 ones are described.

3. The reasons for selecting these four molecules in particular and not others (that could be at least cited to cover the field) should be explained in the section 2 or 3. 

4. The term “non-immunosuppressive” should be defined for clarity. What are the objective, immunological criteria for claiming that a molecule is “non-immunosuppressive”? This description deserves a specific section since it is central in the orientation given to this article, in the choice of the four selected molecules, and also because on an immunological point of view, it is real, fundamental question that is not easy to elaborate.

5. A figure with the four selected molecules could be a plus, particularly if, for each, the cell targets and their mechanisms of action are illustrated.

6. Page 3, line 105: what is the meaning of “relatively safe”? Is this feature acceptable for the Authors?

7. Table 1: Most of the data in human need to be revisited and amended to reflect the present state of the clinical trials that are on-going or finished. For example, in the case of metformin, please update what it is mentioned according to ClinicalTrials.gov Identifier NCT02741960: this is a multicenter, randomised, double-blind placebo controlled trial on the efficacy and safety of add-on metformin to conventional immunosuppressants in SLE. Please correct for the three other molecules and express your conclusion.

8. The references that are quoted are often old. While the field evolves very rapidly, only 14/95 articles that are cited have been published during the two last years (2017 and 2018) and 5 only in 2016.

Author Response

Comments and Suggestions for Authors

Several extensive reviews and monographs have appeared in the recent years on new therapies for SLE, and this represents a lot to read for beginners in the field. With regard to these aspects, this comprehensive review article, which is specially focused on non-immunosuppressive therapies, is welcome. Quite regrettably, however, the presentation of text is sometimes superficial, with typical sentences that can be read in many other reviews and articles (especially in the Introduction) without any critical thinking. The tone that is set remains essentially descriptive and the authors express no opinion or more personal ideas. They do not mention any limitations or criticisms in the analyses (shortly evoked in lines 360-365 of the Conclusion). These aspects are important in a review article because they can orientate future research.   

Response: We have updated our Introduction and Conclusion accordingly. For the Introduction, we have included the reasons why these four molecules were selected, and clarified the usage of the term “non-immunosuppressive” (lines 70 to 76). For the Conclusion, we have included the limitations of current studies and data, and suggestions on how future research may be oriented (lines 384 to 400).

Specific comments

1.       The title is misleading as four so-called “non-immunosuppressive” molecules only are described in this article. This partial choice of selected tools is fully acceptable but this should be mentioned in the title.

Response: The title of the manuscript has been updated to “The Potential Use of Metformin, Dipyridamole, N-acetylcysteine and Statins as Adjunctive Therapy for Systemic Lupus Erythematosus” to better reflect the four drugs described in this paper.

Comments and Suggestions for Authors

Several extensive reviews and monographs have appeared in the recent years on new therapies for SLE, and this represents a lot to read for beginners in the field. With regard to these aspects, this comprehensive review article, which is specially focused on non-immunosuppressive therapies, is welcome. Quite regrettably, however, the presentation of text is sometimes superficial, with typical sentences that can be read in many other reviews and articles (especially in the Introduction) without any critical thinking. The tone that is set remains essentially descriptive and the authors express no opinion or more personal ideas. They do not mention any limitations or criticisms in the analyses (shortly evoked in lines 360-365 of the Conclusion). These aspects are important in a review article because they can orientate future research.   

Response: We have updated our Introduction and Conclusion accordingly. For the Introduction, we have included the reasons why these four molecules were selected, and clarified the usage of the term “non-immunosuppressive” (lines 70 to 76). For the Conclusion, we have included the limitations of current studies and data, and suggestions on how future research may be oriented (lines 384 to 400).

Specific comments

1.       The title is misleading as four so-called “non-immunosuppressive” molecules only are described in this article. This partial choice of selected tools is fully acceptable but this should be mentioned in the title.

Response: The title of the manuscript has been updated to “The Potential Use of Metformin, Dipyridamole, N-acetylcysteine and Statins as Adjunctive Therapy for Systemic Lupus Erythematosus” to better reflect the four drugs described in this paper.

2.       The word “including” (lines 25 and 357) should be deleted since no other molecules than these 4 ones are described.

Response: The word “including” has been updated to the word “namely”.

3.       The reasons for selecting these four molecules in particular and not others (that could be at least cited to cover the field) should be explained in the section 2 or 3. 

Response: We have added an explanation for the choice of these four molecules under the Introduction (lines 70 to 76).

4.       The term “non-immunosuppressive” should be defined for clarity. What are the objective, immunological criteria for claiming that a molecule is “non-immunosuppressive”? This description deserves a specific section since it is central in the orientation given to this article, in the choice of the four selected molecules, and also because on an immunological point of view, it is real, fundamental question that is not easy to elaborate.

Response: We have added an explanation for this under the Introduction (lines 70 to 76). Although there is no official criterion for the term “non-immunosuppressive”, we compared these four drugs (which may modulate the host’s immune system, but without evidence suggesting that they compromise the potency of the host’s immune system) to current immunosuppressive SLE therapy including steroids (which compromise the host’s immune system).

5.       A figure with the four selected molecules could be a plus, particularly if, for each, the cell targets and their mechanisms of action are illustrated.

Response: We have included a new diagram (Figure 2) to summarise the mechanisms of action for each drug. Given that each of the 4 drugs work on multiple different pathways, we have opted to identify the most common pathways in the mechanisms of action of these drugs to be included in this diagram to improve the comprehensiveness of this paper.

6.       Page 3, line 105: what is the meaning of “relatively safe”? Is this feature acceptable for the Authors?

Response: This has been updated to “generally safe”.

7.       Table 1: Most of the data in human need to be revisited and amended to reflect the present state of the clinical trials that are on-going or finished. For example, in the case of metformin, please update what it is mentioned according to ClinicalTrials.gov Identifier NCT02741960: this is a multicenter, randomised, double-blind placebo controlled trial on the efficacy and safety of add-on metformin to conventional immunosuppressants in SLE. Please correct for the three other molecules and express your conclusion.

Response: The type of human studies has been updated in Table 1. For studies which we managed to identify the ClinicalTrials.gov identifier number, we have updated the number and status of the trials accordingly. For all studies, we have opted to leave the studies as “Human study” for consistency since most of the study designs could not be accurately determined.

8.       The references that are quoted are often old. While the field evolves very rapidly, only 14/95 articles that are cited have been published during the two last years (2017 and 2018) and 5 only in 2016.

Response: We had initially concluded our literature search in August 2018 (as stated in section 2: Methods). However, taking this point into consideration, we proceeded to conduct a more up-to-date literature search to ensure that any newer relevant articles published between August 2018 and March 2019 would be accurately included in this paper as well. The updated search yielded multiple new articles but none of them met the inclusion criteria. 

Reviewer 2 Report

It is a well-written and structured review focused on the mechanisms of action, efficacy and side effect of non-immunosuppressive therapeutic agents on SLE, including metformin, dipyridamole, N-acetylcysteine and statins.

I have several suggestions:

- The mechanisms of action of each drug are very complex and for better understanding they should be presented in clarifying schemes.

- It would be advisable to perform a comparative analysis of mechanisms of action and efficacy among the different drugs analyzed.

- It would be interesting to discuss about the feasibility of administering some of these drugs in combination.

- Some previous works, analyzing the beneficial effects of statins on the cardiovascular involvement in SLE patients (i.e. Ruiz-Limon et al., Annals of Rheumatic diseases, 2015) should be analyzed and cited in the review.

Author Response

Comments and Suggestions for Authors

It is a well-written and structured review focused on the mechanisms of action, efficacy and side effect of non-immunosuppressive therapeutic agents on SLE, including metformin, dipyridamole, N-acetylcysteine and statins.

I have several suggestions:

- The mechanisms of action of each drug are very complex and for better understanding they should be presented in clarifying schemes.

Response: We have included a new diagram (Figure 2) as a summary of the mechanisms of actions of each drug to improve the comprehensiveness of this paper.

- It would be advisable to perform a comparative analysis of mechanisms of action and efficacy among the different drugs analyzed.

Response: The newly included Figure 2 aims to give the readers a good comparison of the mechanisms of action among the four drugs analysed.

- It would be interesting to discuss about the feasibility of administering some of these drugs in combination.

Response: We have added a paragraph addressing this in the Conclusion (lines 390 to 400).

- Some previous works, analyzing the beneficial effects of statins on the cardiovascular involvement in SLE patients (i.e. Ruiz-Limon et al., Annals of Rheumatic diseases, 2015) should be analyzed and cited in the review.

Response: We have included a new paragraph with the analysis of the findings of this study under Section 4.1.1 (lines 303 to 306).

Round 2

Reviewer 1 Report

There is a real improvement of the paper. The authors have tried to answer as closely as possible to my questions/remarks and in general their answers are entirely satisfactory. Overall I am satisfied and support the publication of this article.

Reviewer 2 Report

The authors have appropriately raised all my concerns.